# Occupational Health-Related Problems among Portuguese Fitness Instructors

**DOI:** 10.3390/healthcare12090877

**Published:** 2024-04-23

**Authors:** Ricardo Maia Ferreira, Luís Gonçalves Fernandes, Susana Franco, Vera Simões, António Rodrigues Sampaio

**Affiliations:** 1Social Sciences, Education and Sport Department, Polytechnic Institute of Maia, N2i, 4475-690 Maia, Portugal; d011874@ipmaia.pt; 2Physioterapy Department, Polytechnic Institute of Coimbra, Coimbra Health School, 3046-854 Coimbra, Portugal; 3Sport Physical Activity and Health Research & Innovation Center (SPRINT), 4960-320 Melgaço, Portugal; 4Polytechinc Institute of Santarém, School of Sports of Rio Maior, 2040-413 Rio Maior, Portugal; sfranco@esdrm.ipsantarem.pt (S.F.); verasimoes@esdrm.ipsantarem.pt (V.S.); 5Quality of Life Research Center (CIEQV), 2040-400 Santarém, Portugal; 6Research Center in Sports Sciences, Health Sciences and Human Development, CIDESD, University of Maia, 4475-690 Maia, Portugal

**Keywords:** fitness instructors, health, injuries, occupational

## Abstract

Background: The fitness sector has experienced significant expansion, with fitness instructors (FIs) playing a pivotal role. Given the demands of their profession, understanding their health profile is crucial. This study’s purpose is to explore the prevalence of fitness instructors’ occupational health-related problems. Methods: A questionnaire covering sociodemographic, occupational, and health-related items was administered. Statistical analyses, including Mann–Whitney U and chi-square tests, Spearman’s rho correlations, and logistic regressions, were conducted. Results: Fifty-nine FIs reported occupational health-related problems, with the majority occurring during instruction (66.1%), being muscular (32.2%), and knee (15.3%), the most common type and localization. Significant statistical differences were observed between injured and non-injured FIs, including sex (*p* = 0.012), years as an FI (*p* = 0.001), weekly days worked (*p* = 0.039), and daily hours worked (*p* = 0.013). Weak negative (−0.284 – −0.362) statistically significant correlations were found between health problems; weight; height; main activity; and FIs in the workplace. Logistic regressions identified significant models showing that having a sport/physical exercise background and practicing it regularly were less likely to report bursitis (OR 0.018; *p* = 0.020) and hip injuries (OR 0.026; *p* = 0.037). Conclusions: Approximately one-third of FIs reported occupational-related health problems, predominantly musculoskeletal injuries. Sociodemographic, personal, and occupational factors appear to influence the prevalence of these health problems.

## 1. Introduction

Over the past 10–15 years, the fitness industry has shown a rapid increase in the number of fitness clubs, members, and employees [1]. In Europe alone, 63.1 million individuals are health or fitness club members, with an expected annual growth rate of 12.3% [2]. Furthermore, the 63,830 fitness clubs across Europe collectively generate EUR 28 billion in revenue, marking an expected annual growth rate of 0.5% and 66%, respectively [2]. It is estimated that almost 750,000 persons work in this industry in Europe, the USA, and Australia [3]. One key element for client satisfaction and retention, essential for the sector’s growth, is the fitness instructors [4,5].

Fitness instructors are professionals who generally lead, instruct, assess, prescribe, and motivate individuals (or groups) in activities, including different types of exercises and/or classes aiming to enhance fitness and mindfulness [1,3,6]. Fitness instructors work with individual clients (personal trainers) to design, explain, and demonstrate various exercises and routines or with group classes (group training instructors) where they teach, organize, and lead fitness lessons [1]. These classes typically last between 30 and 90 min, where instructors may set the music and choreography to workout routines while often incorporating specific exercise equipment (e.g., stationary bicycles, weights, etc.) and targeting several aspects of physical fitness (e.g., endurance, velocity, strength, agility, flexibility, coordination, balance, etc.) [1].

The fitness instructor profession is physically demanding [1]. Often, fitness instructors find themselves actively participating in exercises in their training sessions, and they may work many hours, days, nights, weekends, and holidays, and may even have to travel to different gyms or clients’ homes, which can lead to a variety of musculoskeletal injuries [1,5]. Furthermore, inherent factors of the profession can also contribute to various health conditions, including respiratory or urinary infections, headaches, vocal problems, and even psychological or eating disorders [1,6].

Despite the numerous health-related benefits associated with physical activity and exercise, the risk of developing occupational health-related problems among fitness instructors appears almost inevitable [7]. However, knowledge regarding these occupational health-related problems among fitness instructors remains limited [1,3]. The situation in Portugal is particularly concerning as, to our knowledge, there has been no epidemiological study conducted on this population, and there is no clear commitment to training and informing workers about health promotion and injury prevention [8]. Decisions often have to be made with poor data (or no data) or data without the proper context [9]. Therefore, the purpose of this study was to explore the prevalence of fitness instructors’ occupational health-related problems through a self-reported questionnaire conducted in a Portuguese sample.

## 2. Materials and Methods

This study followed the Ethical Principles of the Helsinki Declaration [10] and was approved by the Polytechnic Institute of Maia ethics committee. To achieve the objectives, a retrospective, cross-sectional, self-reporting study was conducted, following established methodological guidelines [11,12,13,14].

### 2.1. Participants and Data Collection

Participants were recruited from various fitness centers spanning across all regions of Portugal. These centers employed fitness instructors in both individual (personal training) and group classes. To ensure an accurate population sample, the potential participants were reached by the communication channels of the fitness centers. Fitness managers or directors were requested to contact their colleagues directly for the e-survey sharing. At this time, in addition to requesting the fitness center authorization for the employee’s participation in the study, the authors informed about the study’s procedures, objectives, and eligibility criteria. The questionnaire link was shared with the potential participants in June 2023, and they completed it through the software Google Forms.

Before distributing the e-survey, the questionnaire underwent a thorough development process. This involved reviewing the most recent and relevant literature and conducting pre-tests by the authors and potential target population. The evaluation focused on completion time, question order, syntax, clarity, design, attractiveness, logic, appropriate question types, and response formats. Furthermore, the questionnaire was validated by an external expert panel (of 2 independent and methodological experienced PhDs with exercise and health backgrounds, respectively), where they were able to comment and suggest improvements.

The Raosoft Sample Size Calculator (http://www.raosoft.com/samplesize.html accessed on 27 May 2020) was used to determine the required sample size. The sample size goal was set at 369 responses, based on 9061 active working fitness instructors [15], with a 95% confidence level, a margin of error of 5%, and a response distribution of 50% [16]. To ensure that the sample size goal was achieved, a thank you note and a reminder containing the questionnaire link were sent at two-, four-, and six-week intervals.

The established eligibility criteria for participation in the study were sufficient skills in reading and writing in Portuguese, being over 18 years old, holding a valid professional legal title of exercise technician, and actively working as a fitness instructor.

### 2.2. Questionnaire

Before initiating the e-survey (on the front page), participants were provided with comprehensive information regarding the study’s objectives and context, their data protection rights, and the utilization of results (which would be analyzed anonymously and confidentially, solely for statistical purposes within an academic setting), the criteria for participant selection, the information that no financial (or other incentives) support was provided, clear instructions about how to fill and complete the e-survey, the possibility to stop, review and change their answers at any time, and the e-mail address for possible clarifications. The consent for participation in the study was obtained through an informed consent statement. Upon consenting, the fitness instructors were allowed to initiate the e-survey. The e-survey took 10–15 min to complete and included 25 close-ended questions, divided into 2 main stages (18 occupational and sociodemographic-related questions and 7 health disorders and injuries-related questions):*Occupational and sociodemographic-related items*. This questionnaire phase contained questions regarding age, sex, height, weight, education level, sports background, years as fitness instructors, weekly instruction, workload, main instruction activity, workplace, and other professional activities.*Health disorders and injuries-related items*. In this phase, the fitness instructors self-reported their health disorders and injuries related to their profession by answering the following question: “Have you experienced any health problem related to your occupation as a fitness instructor?” Upon a positive response, responders were asked to specify their frequency, location, type, context, time to return to work, health history, and management. The items in this phase were adapted from a consensus statement regarding injury registration [17]. To ensure a proper questionnaire filling, definitions and examples were given throughout the items, helping to contextualize the readers.

### 2.3. Data Analysis

The data retrieved from the online questionnaire was securely entered into a protected database. Subsequently, the data was organized and presented in tabular and graphical formats using Microsoft Excel (Microsoft Corp, Redmond, Washington, DC, USA) and IBM SPSS 26.0 (International Business Machines Corporation, Statistical Package for the Social Sciences, Armonk, NY, USA) software. Respondents who did not sign the informed consent, declined to participate in the study, or had missing/incomplete data were excluded from the analysis.

Following verification of data normal distribution, two independent samples of non-parametric Mann–Whitney U (ordinal data) and chi-square tests (nominal data) were conducted to compare injured and non-injured fitness instructors. Additionally, Spearman’s rho correlations were performed between all variables. The strength of the correlations was evaluated according to the following criteria [18]: 0 to 0.20—negligible; 0.21 to 0.40—weak; 0.41 to 0.60—moderate; 0.61 to 0.80—strong; and 0.81 to 1.00—very strong. Moreover, logistic regression analyses using the enter model were conducted to examine the associations between the fitness instructors’ characteristics, occupational factors, and health disorders/injuries. Prior to examining the associations between variables, some categories were collapsed and renamed to derive stable models in logistic regression analyses. This decision was informed by previous analysis of the response frequency of items and contextual factors. For sociodemographic and personal-related items, it was adjusted according to the following: age, body weight, educational level, sports background weekly practice, sports background time spent, and sports background competitive level. For the occupational-related items, it was altered: years working as a fitness instructor, weekly days working as a fitness instructor, daily hours working as a fitness instructor, main activity duration, and working at other professions. For the items related to health problems, it was changed according to the following: number of health problems and time returning to work. A significance level of *p* of 0.05 defined whether a model needed to be reported [19]. Odds ratios (ORs) and their 95% confidence intervals (CIs) were determined for each level of the independent variables. Cis, including 1.0, were considered statistically non-significant [20]. The R^2^ values were interpreted as follows [21]: R^2^ < 2%—very weak; 2% ≤ R^2^ < 13%—weak; 13% ≤ R^2^ < 26%—moderate; and R^2^ ≥ 26%—substantial.

## 3. Results

From the estimated 9061 actively working fitness instructors in Portugal, only 1462 could be reached, and from those, 213 showed interest in participating in the study. One participant was excluded due to failure to sign the informed consent, resulting in 212 complete questionnaires. Figure 1 shows the study’s flow diagram.

Among the 212 fitness instructors who completed the questionnaire, there was more preponderance to be males (59.4%), in the 35–39 age group (22.6%), with 50–74 kg in weight (57.1%), and 1.75–2.00 m in height (40.5%). The majority held a bachelor’s degree (44.8%) and had engaged in sports or physical activity before becoming a fitness instructor (96.8%). From those who practiced sports or physical activity, they practiced it three or more times per week (64.2%), during 60–89 min per training (49.5%), at a competitive level (59.9%), were the most frequent responses. Regarding their fitness instructor work-related questions, the majority were in the 1–3 years time range (23.1%), working 4–6 days per week (77.8%), 7–10 h daily (42.5%), having personal training as the main activity (51.4%), spending 30–59 min per training/activity (51.9%), with other 10–15 fitness instructors working in the facilities (31.6%). Although 54.2% stated not having other professions, among those who did, the most commonly reported physical activity involved standing and walking with moderate physical effort (15.1%). Table 1 shows more detailed data.

Out of the 212 fitness instructors who completed the questionnaire, 59 (27.8%) reported experiencing occupational health problems. Among them, 40.7% reported two occupational health problems, occurring mainly during instruction (66.1%). The most prevalent localizations and types were knee injuries (15.3%) and muscular injuries (32.2%). For the knee, tendon injuries were the most common (4–44.4%), and for the muscular injuries, they were reported mainly in the lower back (8–42.1%). These health problems usually had no clinical history (61%) and were managed mostly with the help of a physiotherapist (30.5%), leading to an absenteeism of less than 1 week or 3–4 weeks (both with 27.1%). Table 2 and Figure 2 show the health problems characterization.

In comparison with the fitness instructors who reported having an occupational health problem versus those who reported it was found significant statistical differences in the sex (*p* = 0.012) factor, with females showing a higher propensity for injuries. Additionally, the years as fitness instructors (*p* = 0.001) were an important factor, with those with moderate or high experience being more prone to injuries. Similar results were observed for workload, where the higher workload was associated with increased injury reporting (weekly days working (*p* = 0.039) and daily hours working (*p* = 0.013)). All other variables did not show significant statistical differences. For more information, see Table 1.

Statistical correlations between the health problems and the other variables were also found. One moderate positive (0.496) high statistically significant (*p* ≤ 0.01) correlation was found between the health problems localization and weekly days working as fitness instructors variables. Also, a weak positive (0.346) high statistically significant (*p* ≤ 0.01) correlation was found between the health problem management and main activity as fitness instructors variables. Additionally, a moderate negative (−0.463) high statistically significant (*p* ≤ 0.01) correlation was found between the health problems localization and health problems type variables. A weak negative (−0.362) high statistically significant (*p* ≤ 0.01) correlation was found between the number of health problems and weight variables. Weak positive (0.258–0.273) low statistically significant (*p* ≤ 0.05) correlations were found between the variables: Health problem type vs. Sex; Return-to-work time vs. Years as fitness instructors; Return-to-work vs. Number of health problems. Furthermore, weak negative (−0.258–−0.381) low statistically significant (*p* ≤ 0.05) correlations were found between the variables: Number of health problems vs. Height; Number of health problems vs. Main activity type; Number of health problems vs. Number of fitness instructors at the workplace; Health problem type vs. Height; Health problem type vs. Having sports background; Health problem type vs. Sports competitive level; Health problem type vs. Main activity duration; Health problem localization vs. Height; Health problem management vs. Return-to-work time.

The other non-health problem-related items also had statistically significant correlations. Two very strong, positive, high statistically significant (*p* ≤ 0.01) correlations were found between height and weight (0.817) and having other professions and other professions activity level (0.916). Moreover, strong positive (0.629–0.735) high statistically significant (*p* ≤ 0.01) correlations were found between the variables: Weight vs. Sex; Height vs. Sex; Sports background competitive level vs. Sports activity practice duration; and Years as fitness instructors vs. Age. Additionally, moderately positive (0.499–0.432) high statistically significant (*p* ≤ 0.01) correlations were found between the variables: Sport activity practice duration vs. Sport activity weekly practice; Sport competitive level vs. Sport activity practice duration; Daily working hours vs. Weekly working days; Main activity type vs. Sex; Number of fitness instructors at workplace vs. Main activity as fitness instructors. Moderate negative (−0.417–−0.525) high statistically significant (*p* ≤ 0.01) correlations were found between the variables: Main activity as fitness instructors vs. Age; Main activity as fitness instructors vs. Years as fitness instructors; Number of fitness instructors at the workplace vs. Age. Two weak positive high statistically significant (*p* ≤ 0.01) correlations were found between sport weekly practice and sports background (0.374) and sport competitive level and sports background (0.361) variables. A weak negative high statistically significant (*p* ≤ 0.01) correlation was found between the number of fitness instructors at the workplace and years as fitness instructors variables (−0.350). Also, weakly positive (0.326–0.274) low statistically significant (*p* ≤ 0.05) correlations were found between the variables: Sports activity practice duration vs. Sports background; Sports competitive level vs. Educational level; Main activity duration vs. Sports competitive level; and Having other professions vs. Main activity duration. Two weak negative, low, statistically significant (*p* ≤ 0.05) correlations were found between having other professions and age (−0.298) and other professions’ activity level and age (−0.323). Table 3 shows the detailed information.

Regarding the logistic regressions, two statistically significant models were found. A substantial model was found for bursitis (*p* = 0.020; R^2^ = 29%), wherein individuals with a sports background were less likely to report it compared to those without such background (OR 0.018, 95% CI [0.001–0.538]). Another moderate model was found for injuries in the hip (*p* = 0.050; R^2^ = 25%), whereby individuals engaging in sports activities 2–3 times per week (OR 0.286, 95% CI [0.014–5.660]; *p* = 0.411) or more than 3 times per week (OR 0.026, 95% CI [0.001–0.796]; *p* = 0.037) were less likely to report injuries compared to those without a sports background. Further detailed information is provided in Table 4.

## 4. Discussion

In this study, health problems’ types and localizations stood out, as well as significant personal, sociodemographic, and occupational characteristics that may influence the reporting of health-related disorders.

Regarding the musculoskeletal-related injuries, it appears that muscular (32.2%), tendon (13.6%), bone (8.5%), and joint (6.8%) were the most injury types found in either the knee (15.3%), lower back (13.6%), hip (10.2%), lumbar spine (6.8%), or neck (6.8%). Among these, lower back muscular injuries were the most prevalent (13.6%). These findings align with previous fitness instructors’ occupational health-related problems reported studies [1,3,22,23]. For instance, du Toit and Smith [22] found that aerobics instructors reported a 77% injury rate in the lower limbs, where the leg was the most common injury site (52.9%), followed by the foot/ankle (32.8%), and the knee (20%). In a survey with kickboxing instructors, Romaine et al. [23] found that the injuries were mainly in the back (20%), knee (19%), hip (12%), shoulder (10%), and thigh (8%). Within Norwegian fitness instructors, injuries in the lower leg (29%), knee (15%), ankle (15%), shoulder (12%), lower back (10%), and foot (7%) were mostly found [3]. Italian fitness instructors also reported similar patterns, where the most common injuries were reported in the lower back, ankle, and knee [1].

Hazardous work is an occupation in a dirty, and/or difficult, and/or dangerous environment, which may pose risks of injury, illness, or death [24]. An occupational injury may be defined as one that is caused or made worse by exposure at work [25]. These injuries are generally multifactorial, with different risk factors contributing to their occurrence, such as [26] physical, organizational, psychosocial, individual, and sociocultural. Specifically, for the fitness instructors’ working environment, the risk factors mostly related are [8] physically demanding or painful positions, lifting or moving people, carrying and handling heavy loads, repeated movements, prolonged standing, being subjected to loud noises, using dangerous machinery and work equipment, being exposed to high temperatures, working conditions with poor air quality or inhaling toxic products, and being subjected to high psychological pressures.

Most of the occupational disorders in all workers’ categories are musculoskeletal-related, occasioned by overexertion, heavy loads, or repetitive motions [8,26]. It should be taken into consideration that the disorders found in this and other international studies with fitness instructors’ samples may originate from overuse injuries resulting from repetitive load applied to a tissue [1,17]. Given the nature of their workplace environment, such injuries are expected among fitness instructors [3]. Generally, in addition to all work-related tasks, fitness instructors have to instruct the same time-consuming classes/exercises several times per day/week with a hard exercise physical intensity [1]. A study involving Portuguese fitness instructors revealed an average weekly workload of 39.8 h, with a daily workload of 10 h and 35 min [27]. Additionally, this study found that the fitness instructors reported classes with the most common durations of 30–59 min. It was found that full-time fitness instructors (more than 4–6 weekly days and 7–10 daily hours working) have a higher risk of suffering from a health occupational disorder in comparison with their part-time peers (*p* = 0.039 and *p* = 0.013, respectively). This observation underscores the potential association between prolonged exposure to a specific activity without sufficient recovery time and the likelihood of developing health disorders [28]. Nevertheless, it is also expected that the more experienced a fitness instructor is, the more adaptations will have to an activity, leading to a protective effect [29,30,31,32,33]. However, it seems that this chronic protective effect is not found in this population/profession. Although the exact reasons for this phenomenon remain unclear, reflecting on the results, some workplace factors, such as activity type and duration (water-based vs. land-based—*r* = −0.306; short vs. long—*r* = −0.381) and the number of fitness instructors working at the facilities (*r* = −0.284), may be the answer. Furthermore, an interesting sociodemographic factor emerges regarding the reported injuries and sport/physical exercise background. In this study, it seems that having a sport/physical exercise background could have a protective effect on some health problem types and localizations. In the two statistically significant logistic models, those who had a sport/physical exercise background and practiced it regularly were less likely to report bursitis (OR 0.018 (95% CI [0.001–0.538]; *p* = 0.020) and hip injuries (OR 0.026 (95% CI [0.001–0.796]; *p* = 0.037). Therefore, further exploration is warranted to elucidate the factors underlying the protective effects of exercise within this profession.

Other health disorders, such as respiratory infections, vocal problems, and physiological disorders, were also reported. Apparently, these are common health disorders related to the fitness instructing profession, as was also reported in other similar studies [1,6]. It is reasonable to associate these disorders with the typical demands of the job [1]. Fitness instructors in their classes require loud verbal instructions/motivations while performing exercises (due to big spaces, and/or a large number of members attending the classes, and/or loud background music), thereby making the control of breathing and airflow movement more stressful, frequently leading to aphonia or other vocal problems [1,34,35,36,37,38,39,40,41,42,43,44]. Respiratory infections could be related to this since they could lead to voice loss [36]. However, in this case, it is expected that respiratory infections could be more linked to other occupational factors, such as working closely/directly with other people, with poor air quality/ventilation, in hot and humid places [1,45,46].

One health problem that unexpectedly demonstrated to be prevalent was psychological disorders. In fact, in this study, 2.36% (8.5% of the health problems) reported having a psychological disorder. This could be related to the professional demands as, on the one hand, fitness instructors are not fully satisfied with their job and, on the other hand, they are considered role models (either based on their image or healthy lifestyle), creating a psychological “pressure”, making it easier to develop symptoms of anxiety, depression, and eating disorders [5,6]. These symptoms are a predictor of both instruction-related injuries and musculoskeletal pain [3]. Moreover, the country is coming to a new reality in the fitness industry. Portugal has maintained a pace of growth in the fitness industry compared to other countries. The results have been consistent since 2017, reaching the maximum point of revenues (EUR 289 million), clubs (1100), number of members (688,210), and a market penetration rate of 6.7% in 2019 [15]. After a drop in these indicators in 2020 (due to the social impact of COVID-19), there was a stabilization in 2021, and in 2022, an upturn in the industry was registered with clubs earning around EUR 230 million, 800 clubs, more than 691,000 members, and a market penetration rate of 6.7% [15]. However, there has been a noticeable decrease in the number of fitness instructors over the same period, where in 2018, there were 12,872; in 2019, 12,086; in 2020, 9822; in 2021, 9652; and in 2022, 9061 [15]. Fewer instructors and an increasing number of members over the years can lead to greater psychological stress on the instructor by having to interact with more members, tighter time management between members/exercises/classes, and instruct more people simultaneously at the workplace (the member/instructor ratio in 2018 was 46, and in 2022 was 76). Additional workplace environment characteristics associated with psychological disorders found in other populations can also be suitable to our sample, such as [8] working with strict deadlines, not taking the recommended work breaks, and having to deal with irritable/difficult customers. Furthermore, although in this sample, most of the fitness instructors reported not having an additional job (54.2%), other nationwide studies reveal that only 31% of fitness instructors work full-time, and 55% work 30 h/week or less [15], having to reconcile the career with another job (37.7%) [27]. This may cause limited free time, a constant stressful thought of losing their job, and difficult work–family–social life management, leading to greater psychological pressure [3,8].

Psychological disorders are a special concern among female instructors as they are more likely to develop such problems caused by acts of discrimination, physical violence, sexual or moral harassment, and disturbances in image or personality [6,8]. In our study, sex is one of the most important characteristics, where statistically significant differences (*p* = 0.012) were found between those who reported having health disorders and those who did not; being female is a risk factor. In the professionals who reported having a health problem, women were the majority (54.2%), unlike those without injuries or even in the overall sample (35.3% and 40.6%, respectively). This could be explained by the overall poor quality of life of female fitness instructors in comparison with their male peers. A study [47] found significant differences in the physical, psychological, and environmental domains, suggesting that lower levels of job satisfaction, workplace sexual harassment, psychological stress, social trends, and specific health-related situations were the main factors. Moreover, when considering anthropometric factors such as weight and height, further insights emerge. Statistically significant correlations were found between weight, height, and the number of reported injuries. Fitness instructors who were smaller in height and weight were more likely to have a higher number of injuries (−0.286 and −0.362, respectively). Given the tendency for women to be lighter and shorter in comparison with men, this pattern is further underscored. In this study, this tendency was also demonstrated through statistically significant correlations (*p* ≤ 0.01; *n* = 212; 0.579 and 0.587, respectively). Although females are not the most predominant sex in this study, it is important to give further special attention to this characteristic because, taking into account other Portuguese studies, the differences between sexes are getting nearer [5,27,47,48,49,50].

Furthermore, although no relation was found in the age factor, this could also be an important influential factor in the health disorders emerging. A statistically significant correlation was found between age and years of working as a fitness instructor (0.735; *p* ≤ 0.01), and it seems that years of working as a fitness instructor may be an important factor, as the more years of working as a fitness instructor the more likely to have a health occupational disorder (*p* = 0.001). In other professions, this association is also found [8]. The justification appears to be identical to the one explored earlier on workload factors, where the longer the exposure to an activity, the greater the predisposition to report a musculoskeletal injury. Other health problems, namely psychological stress, can also be associated with this factor, as other studies with Portuguese fitness instructors have shown that workload tends to increase proportionally with age [27]. Therefore, age should also be a factor of special attention in this profession.

## 5. Limitations

One of the limitations of this study is the fact that the minimum number of participants to have a representative sample of the population was not reached, thus limiting the appreciation of the results. The second major limitation relates to the questionnaire operationalization. First, it was not sensitive enough to cover all health problems (in the closed answers, when the “other” was selected, it was not specifically known what it was). Second, due to the limitations of the questionnaire software, respondents could only explore one health problem (the most important/severe), thus not having the perception of all the health problems that fitness instructors may have suffered throughout their careers. A third and final limitation is that since the questionnaire is retrospective and self-reporting, responses to health-related items may have reporting bias due to the participants’ lack of recall and specific health literacy/knowledge.

## 6. Conclusions

In conclusion, almost one-third of fitness instructors may suffer at least one occupation-related health problem in their careers. Among these, musculoskeletal injuries emerge as the most prevalent, particularly affecting muscles, tendons, bones, and joints in areas such as the knee, lower back, hip, lumbar spine, or neck. Sociodemographic, personal, and occupational characteristics may influence work-related health problems, especially sex, anthropometric measures, workload, years as fitness instructors, main activity, and sports/physical exercise background. Taking into consideration the fitness instructors’ work environment and the occupational disorders reported, it would be necessary to pay close attention and implement appropriate legal actions to protect this population since it has the conditions to be considered a hazardous profession.

## Figures and Tables

**Figure 1 healthcare-12-00877-f001:**
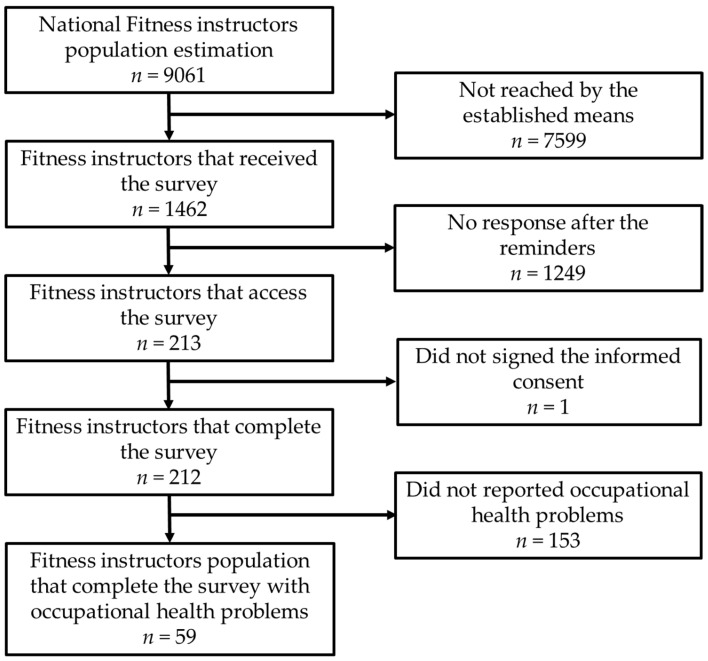
Questionnaire views, participation, and completion.

**Figure 2 healthcare-12-00877-f002:**
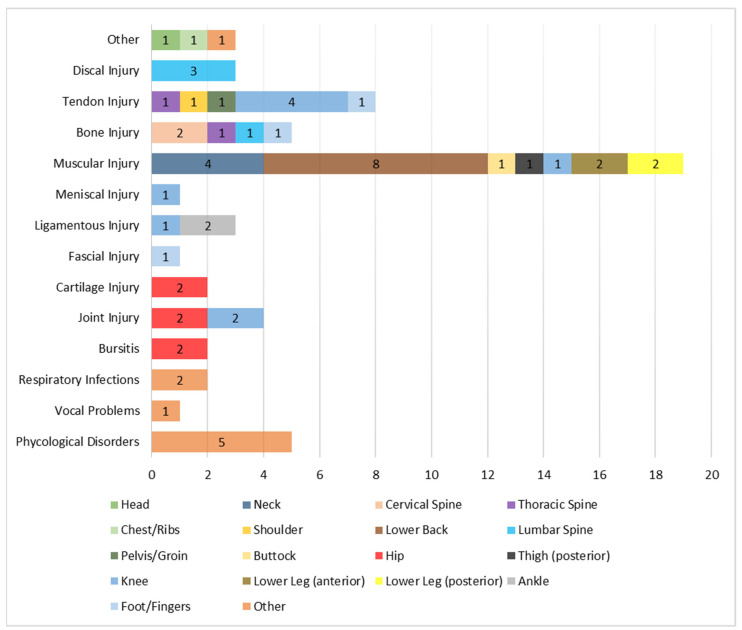
Reported health problems type and localization.

**Table 1 healthcare-12-00877-t001:** Fitness instructors’ personal and occupational characteristics.

Variable	Total (*n* = 212 (100%)) *n* (%)	Unreported Health Problems (*n* = 153 (72.2%)) *n* (%)	Reported Health Problems (*n* = 59 (27.8%)) *n* (%)	Statistics (*p*) Health Problems vs. No Health Problems
Age group (years)				0.390 *
*18–24*	32 (15.1)	27 (17.6)	5 (8.5)	
*25–29*	46 (21.7)	36 (23.5)	10 (16.9)	
*30–34*	42 (19.8)	24 (15.7)	18 (30.5)	
*35–39*	48 (22.6)	32 (20.9)	16 (27.1)	
*40–44*	26 (12.3)	19 (12.4)	7 (11.9)	
*45–50*	18 (8.5)	15 (9.8)	3 (5.1)	
Sex				**0.012 ****
*Female*	86 (40.6)	54 (35.3)	32 (54.2)	
*Male*	126 (59.4)	99 (64.7)	27 (45.8)	
Body weight (kilograms)				0.621 *
*<50*	4 (1.9)	4 (2.6)	0 (0)	
*50–74*	121 (57.1)	84 (54.9)	37 (62.7)	
*75–100*	82 (38.7)	61 (39.9)	21 (35.6)	
*>100*	5 (2.4)	4 (2.6)	1 (1.7)	
Body height (meters)				0.393 *
*−1.50*	1 (0.5)	1 (0.7)	0 (0)	
*1.50–1.74*	125 (59)	87 (56.9)	38 (64.4)	
*1.75–2.00*	86 (40.5)	65 (42.5)	21 (35.6)	
Educational level				0.452 *
*Physical Exercise Technical Expert (Undergraduate)*	48 (22.6)	40 (26.1)	8 (13.6)	
*Bachelor*	95 (44.8)	61 (39.9)	34 (57.6)	
*Master*	62 (29.2)	49 (32)	13 (22)	
*PhD*	2 (0.9)	1 (0.7)	1 (1.7)	
*Other*	5 (2.4)	2 (1.3)	3 (5.1)	
Sport or physical activity background				0.856 **
*Yes*	204 (96.8)	147 (96.1)	57 (96.6)	
*No*	8 (3.2)	6 (3.9)	2 (3.4)	
Sport or physical activity weekly practice				0.958 *
*Once per week*	2 (0.9)	2 (1.3)	0 (0)	
*2 times per week*	17 (8)	10 (6.5)	7 (11.9)	
*3 times per week*	49 (23.1)	38 (24.8)	11 (18.6)	
*More than 3 times a week*	136 (64.2)	97 (63.4)	39 (66.1)	
*No sport or physical activity*	8 (3.8)	8 (3.9)	2 (3.4)	
Sport or physical activity practice (minutes)				0.101 *
*30–59*	44 (20.8)	31 (20.3)	13 (22)	
*60–89*	105 (49.5)	84 (54.9)	21 (35.6)	
*90–120*	40 (18.9)	26 (17)	14 (23.7)	
*>120*	15 (7.1)	6 (3.9)	9 (15.3)	
*No sport or physical activity*	8 (3.8)	6 (3.9)	2 (3.4)	
Sport or physical activity at competitive level				0.643 *
*Recreational*	62 (29.2)	42 (27.5)	20 (33.9)	
*Beginner*	15 (7.1)	13 (8.5)	2 (3.4)	
*Competitive*	127 (59.9)	92 (8.5)	35 (59.3)	
*No sport or physical activity*	8 (3.8)	6 (3.9)	2 (3.4)	
Years as fitness instructor				**0.001 ***
*<1*	25 (11.8)	23 (15)	2 (3.4)	
*1–3*	49 (23.1)	38 (24.8)	11 (18.6)	
*4–6*	31 (14.6)	25 (16.3)	6 (10.2)	
*7–9*	40 (18.9)	26 (17)	14 (23.7)	
*10–13*	27 (12.7)	18 (11.8)	9 (15.3)	
*14–16*	19 (9)	12 (7.8)	7 (11.9)	
*>20*	21 (9.9)	11 (7.2)	10 (16.9)	
Weekly days working as fitness instructor				**0.039 ***
*1*	2 (0.9)	2 (1.3)	0 (0)	
*2–3*	15 (7.1)	14 (9.2)	1 (1.7)	
*4–6*	165 (77.8)	118 (77.1)	47 (79.7)	
*7*	30 (14.2)	19 (12.4)	11 (18.6)	
Fitness instructor daily working (hours)				**0.013 ***
*<1*	2 (0.9)	2 (1.3)	0 (0)	
*1–3*	27 (12.7)	22 (14.4)	5 (8.5)	
*4–6*	60 (28.3)	51 (33.3)	9 (15.3)	
*7–10*	90 (42.5)	54 (35.3)	36 (61)	
*>10*	33 (15.6)	24 (15.7)	9 (15.3)	
Fitness instructor’s main activity				0.114 **
*Aquafitness classes*	31 (14.6)	17 (11.1)	14 (23.7)	
*Group classes*	46 (21.7)	31 (20.3)	15 (25.4)	
*Exercise room monitoring*	20 (9.4)	16 (10.5)	4 (6.8)	
*Personalized training*	109 (51.4)	84 (54.9)	25 (42.4)	
*Mind and body classes*	6 (2.8)	5 (3.3)	1 (1.7)	
Main activity duration (minutes)				0.624 *
*<30*	2 (0.9)	2 (1.3)	0 (0)	
*30–59*	110 (51.9)	80 (52.3)	30 (50.8)	
*60–89*	21 (9.9)	15 (9.8)	6 (10.2)	
*90–120*	19 (9)	14 (9.2)	5 (8.5)	
*>120*	60 (28.3)	42 (27.5)	18 (30.5)	
More Fitness instructors in the workplace?				0.348 *
*Yes, <5*	30 (14.2)	17 (11.1)	13 (22)	
*Yes, 5–9*	66 (31.1)	50 (32.7)	16 (27.1)	
*Yes, 10–15*	67 (31.6)	51 (33.3)	16 (27.1)	
*Yes, >15*	48 (22.6)	34 (22.2)	14 (23.7)	
*No*	1 (0.5)	1 (0.7)	0 (0)	
Other profession(s) besides Fitness instructor?				0.379 *
*Yes, 1*	73 (34.4)	55 (35.9)	18 (30.5)	
*Yes, 2*	22 (10.4)	16 (10.5)	6 (10.2)	
*Yes, 3 or more*	2 (0.9)	2 (1.3)	0 (0)	
*No*	115 (54.2)	80 (52.3)	35 (59.3)	
Other profession(s) physical activity level				0.311 *
*Sedentary*	7 (3.3)	6 (3.9)	1 (1.7)	
*Sitting and walking, without physical efforts*	14 (6.6)	11 (7.2)	3 (5.1)	
*Sitting and walking, with moderate physical efforts*	14 (6.6)	10 (6.5)	4 (6.8)	
*Sitting and walking, with heavy physical efforts*	1 (0.5)	1 (0.7)	0 (0)	
*Standing and walking, without physical efforts*	18 (8.5)	13 (8.5)	5 (27.8)	
*Standing and walking, with moderate physical efforts*	32 (15.1)	24 (15.7)	8 (13.6)	
*Standing and walking, with heavy physical efforts*	11 (5.2)	8 (5.2)	3 (5.1)	
*No other professional activities*	115 (54.2)	80 (52.3)	35 (59.3)	

Note: Bold—significant statistic differences; * Mann–Whitney U test; ** Chi-square test; Kg—kilograms; m—meters; min—minutes; h—hours.

**Table 2 healthcare-12-00877-t002:** Fitness instructors reported health problems characterization (*n* = 59).

Variable	*n* (%)
N° health problems	
*1*	15 (25.4)
*2*	24 (40.7)
*3*	15 (25.4)
*4*	2 (3.4)
*5*	1 (1.7)
*>5*	2 (3.4)
Health problem localization	
*Head*	1 (1.7)
*Neck*	4 (6.8)
*Cervical spine*	2 (3.4)
*Thoracic spine*	2 (3.4)
*Chest/Ribs*	1 (1.7)
*Shoulder*	1 (1.7)
*Lower back*	8 (13.6)
*Lumbar spine*	4 (6.8)
*Pelvis/Groin*	1 (1.7)
*Buttock*	1 (1.7)
*Hip*	6 (10.2)
*Thigh (posterior)*	1 (1.7)
*Knee*	9 (15.3)
*Lower leg (anterior)*	2 (3.4)
*Lower leg (posterior)*	2 (3.4)
*Ankle*	2 (3.4)
*Foot/Fingers*	3 (5.1)
*Other*	9 (15.3)
Health problem type	
*Psychological disorders*	5 (8.5)
*Vocal problems*	1 (1.7)
*Respiratory infections*	2 (3.4)
*Bursitis*	2 (3.4)
*Joint injury*	4 (6.8)
*Cartilage injury*	2 (3.4)
*Fascial injury*	1 (1.7)
*Ligamentous injury*	3 (5.1)
*Meniscal injury*	1 (1.7)
*Muscular injury*	19 (32.2)
*Bone injury*	5 (8.5)
*Tendon injury*	8 (13.6)
*Discal Injury*	3 (5.1)
*Other*	3 (5.1)
Occupational activity leading to health problems	
*During instruction*	39 (66.1)
*After instruction*	13 (22)
*Other*	7 (11.9)
Return to work duration	
*<1 week*	16 (27.1)
*1–2 weeks*	10 (16.9)
*3–4 weeks*	16 (27.1)
*1–3 months*	8 (13.6)
*4–6 months*	6 (10.2)
*7–12 months*	2 (3.4)
*>1 year*	1 (1.7)
Health problem history	
*Recurrence*	23 (39)
*First time*	36 (61)
Health problem management	
*Active self-management*	2 (3.4)
*Self-medication/Supplementation*	2 (3.4)
*Physiotherapist*	18 (30.5)
*Physician (surgery)*	5 (8.5)
*Physician (injection)*	1 (1.7)
*Physician (medication)*	15 (25.4)
*Non-conventional medicine*	5 (8.5)
*Rest*	6 (10.2)
*No intervention*	2 (3.4)
*Other*	3 (5.1)

**Table 3 healthcare-12-00877-t003:** Spearman correlations between the personal, occupational, and health problems variables (*n* = 59).

Variables	Age	Sex	Weight	Height	Education	Sport Background	Sport Weekly Practice	Time at Sport Activity	Sport Level	Years as FI	Weekly Working Days	Working Hours	Main Activity Type	Main Activity Duration	FIs at Workplace	Other Professions	Other Professions’ Activity Level	Health Problems *n*	Health Problem Type	Health Problem Localization	Occupational Activity Leading to the Health Problem	Return to Work	Health Problem History
Sex	−0.108																						
Weight	0.000	**0.629 ****																					
Height	0.151	**0.667 ****	**0.817 ****																				
Education	0.023	0.142	0.065	0.156																			
Sport Background	−0.221	0.016	0.046	0.056	0.089																		
Sport weekly practice	−0.098	0.244	0.196	0.197	0.233	**0.374 ****																	
Time at sport activity	0.093	0.054	0.037	0.132	0.188	**0.326 ***	**0.491 ****																
Sport level	0.143	0.146	0.040	0.079	**0.310 ***	**0.361 ****	**0.499 ****	**0.635 ****															
Years as FI	**0.735 ****	−0.157	−0.035	0.084	0.081	−0.198	−0.113	0.044	0.116														
Weekly working days	−0.029	−0.137	−0.072	−0.140	0.067	−0.079	0.146	−0.011	0.023	−0.018													
Working hours	**−0.306 ***	0.190	0.181	0.071	0.084	0.031	0.106	−0.088	−0.031	−0.118	**0.431 ****												
Main activity type	**−0.525 ****	**0.432 ****	0.199	0.218	0.029	0.047	0.053	−0.105	−0.089	**−0.417 ****	−0.025	0.149											
Main activity duration	−0.120	−0.188	−0.069	−0.079	0.071	0.246	0.078	0.128	**0.274 ***	−0.074	−0.042	0.115	0.081										
FIs at workplace	**−0.491 ****	0.216	0.185	0.127	0.090	−0.006	0.205	−0.123	−0.067	**−0.350 ****	−0.009	0.046	**0.447 ****	0.140									
Other professions	**−0.298 ***	0.053	−0.028	−0.088	−0.039	0.151	0.080	−0.012	−0.083	−0.175	−0.004	0.083	0.091	0.050	**0.304 ***								
Other professions’ activity level	**−0.323 ***	0.032	−0.055	−0.170	−0.054	0.149	0.169	0.020	−0.045	−0.142	0.001	0.156	0.077	0.088	0.230	**0.916 ****							
N° of health problems	0.170	−0.244	**−0.362 ****	**−0.286 ***	−0.137	−0.029	−0.006	0.057	−0.037	0.102	0.151	0.049	**−0.306 ***	−0.024	**−0.284 ***	0.037	0.100						
Health problem type	0.071	**0.258 ***	0.237	**0.287 ***	−0.077	**−0.258 ***	0.182	−0.231	**−0.281 ***	−0.066	−0.165	0.026	−0.053	**−0.381 ****	0.024	−0.053	−0.064	0.106					
Health problem localization	−0.177	−0.097	−0.125	**−0.274 ***	−0.073	0.030	0.036	0.013	0.163	−0.119	**0.496 ****	0.186	0.177	0.195	0.082	−0.007	0.048	0.005	**−0.463 ****				
Occupational activity leading to health problem	−0.039	0.172	0.097	0.109	−0.103	0.039	0.085	0.093	0.042	−0.069	0.0211	−0.080	0.137	−0.098	0.087	0.236	0.153	−0.209	0.073	0.183			
Return to work	0.127	−0.133	−0.096	0.000	0.136	0.045	0.123	0.087	0.099	**0.273 ***	−0.200	−0.169	−0.121	0.131	−0.142	−0.072	−0.013	**0.273 ***	0.036	−0.009	−0.099		
Health problem history	0.009	−0.103	0.192	0.086	−0.043	−0.042	0.055	−0.044	0.014	0.028	−0.018	−0.138	−0,065	−0.108	−0.128	0.002	0.043	−0.253	−0.118	0.220	−0.044	0.159	
Health problem management	0.183	0.059	0.054	0.027	−0.001	−0.059	−0.188	−0.170	−0.101	0.038	0.158	0.032	0.078	−0.043	−0.006	0.025	−0.020	−0.016	−0.014	0.072	**0.346 ****	**−0.309 ***	−0.161

Note: Bold—significant statistical correlations; ** *p* ≤ 0.01; * *p* ≤ 0.05; FI, fitness instructor.

**Table 4 healthcare-12-00877-t004:** Spearman correlations between the personal, occupational, and health problems variables.

Injury (Present)	Factor Level	Odds Ratio (95% CI)	*p*	*R* ^2 a^
Bursitis				
	*Sport Background*		*0.020*	*0.294*
	Yes	0.018 [0.001; 0.538]		
	No	Reference ^b^		
Hip				
	*Sports background weekly practice*		*0.050*	*0.252*
	2–3 times per week	0.286 [0.014; 5.660]	0.411	
	More than 3 times per week	0.026 [0.001; 0.796]	0.037	
	Did not practice	Reference ^b^		

Note: ^a^ Nagelkerke R^2^; ^b^ In logistic regression, one level of the independent variable serves as reference against which the odds of the other levels occurring are determined.

## Data Availability

The data presented in this study are available upon request from the corresponding author, RMF.

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
