# Peer review of "Occupational Health-Related Problems among Portuguese Fitness Instructors"

_healthcare, 2024, doi:10.3390/healthcare12090877_

Round 1

Reviewer 1 Report

Comments and Suggestions for Authors

Review of Occupational Health-Related Problems Among Portuguese Fitness Instructors

The structure of this paper is commendably organized. It presents a concise yet comprehensive rationale for the study, although the discussion on potential injuries related to the specific nature of fitness work could be expanded. The justification for the chosen survey method and a brief overview of the validation process by two experts are well-articulated. However, the paper's notable shortfall is the absence of references to other validated studies on pain and related issues. It could be argued that the authors should publish a full validation study of the newly developed survey in prestigious journals before its application. To counter this, the authors could enhance their defense by elaborating on the similarities between their work and existing literature on injury-related research, as their survey appears to mirror others closely.

The authors conduct a standard series of tests, covering all aspects up to regression models. Upon examining the data, I noticed the omission of a current trend in data sharing—making research data available online. Given the EU's adherence to FAIR and open data policies, and considering MDPI's data hosting services, this oversight is significant. I am keen on scrutinizing your data personally, as I have identified certain issues with how it is presented:

  • The reported number of injuries and health problems is relatively low. Given the reduced sample size, employing data visualization techniques, such as a frequency count table, would provide a clearer context of the injuries.
  • It should be possible to precisely identify the setups that are most burdensome.
  • The methodology for converting some variables for model analysis is unclear. The categorization of fitness activities as either work or individual sports activities, and the summation of loads for legs as a factor, are confusing. Additionally, without a contingency table, it is difficult to ascertain the amount of time these physically active individuals spend as fitness instructors.
  • In conclusion, I would recommend employing clustering to create specific profiles in such analyses. The overlap in data, due to the difficulty in distinguishing between work and sport activities within this particular group and survey, complicates the analysis. Furthermore, the question of whether instructors in mind and body classes, who are less likely to report injuries compared to those who constantly train with clients or personal trainers, are included in the injury-reporting group remains unanswered.

The use of classical statistics in reporting injury-related outcomes in empirical studies can be challenging, leading to numerous uncertainties and raising many questions. While the discussion section is well-executed, there might be an inherent bias in your results.

I recommend a revision of your work or a request for clarification on why my suggested approach may be incorrect—although this perspective is based on the presentation of your text. Therefore, at a minimum, further explanations should be provided. However, I strongly suggest considering a different descriptive statistical approach, employing contingency tables or clustering methods. Your statistical software should support these options.

Author Response

The structure of this paper is commendably organized. It presents a concise yet comprehensive rationale for the study, although the discussion on potential injuries related to the specific nature of fitness work could be expanded. The justification for the chosen survey method and a brief overview of the validation process by two experts are well-articulated. However, the paper's notable shortfall is the absence of references to other validated studies on pain and related issues. It could be argued that the authors should publish a full validation study of the newly developed survey in prestigious journals before its application. To counter this, the authors could enhance their defense by elaborating on the similarities between their work and existing literature on injury-related research, as their survey appears to mirror others closely.

Response: Thank you for your feedback. We strive to gather comprehensive information on health issues in fitness instructors from available literature. However, due to the limited research in this area, a complete understanding of the problem is still lacking. Therefore, we have gathered and contextualize data with our findings by drawing parallels with related professions such as martial arts (e.g., Romaine, L.J., Davis, S. E., Casebolt, K., and K.A. Harrison, Incidence of injury in kickboxing participation. The Journal of Strength & Conditioning Research, 2003. 17(3): p. 580-586), swimming instructors (e.g., Merati, G., Bonato, M., Agnello, L., Grevers, D., Gunga, H.C., Mendt, S., and M.A. Maggioni, Occupational Disorders, Daily Workload, and Fitness Levels Among Fitness and Swimming Instructors. Frontiers in public health, 2021. 9: p. 666019), and fitness class instructors (e.g., du Toit, V. and R. Smith, Survey of the effects of aerobic dance on the lower extremity in aerobic instructors. Journal of the American Podiatric Medical Association, 2001. 91(10): p. 528-532; Mikhail, L., Venkatraman, A., Dahlby, H., and S. Khosla, Are indoor cycling instructors riding their way to vocal injury? acute effects of a single class on measures of voice production. Journal of Voice, 2022. 36(5): p. 695-700). Additionally, we have provided insights from studies exploring the profession within the Portuguese context to enrich the discussion (e.g., Ramos, L.R., Esteves, D., Vieira, I., Franco, S., and V. Simões, Job satisfaction of fitness professionals in Portugal: a comparative study of gender, age, professional experience, professional title, and educational qualifications. Frontiers in Psychology, 2021. 11: p. 621526). Also, we compared and discuss, our data with other studies that explored specifically the fitness instructor profession in pain and other related issues (e.g., Bratland-Sanda, S., Sundgot-Borgen, J., and G. Myklebust, Injuries and musculoskeletal pain among Norwegian group fitness instructors. European journal of sport science, 2015. 15(8): p. 784-792.).

The authors conduct a standard series of tests, covering all aspects up to regression models. Upon examining the data, I noticed the omission of a current trend in data sharing—making research data available online. Given the EU's adherence to FAIR and open data policies, and considering MDPI's data hosting services, this oversight is significant. I am keen on scrutinizing your data personally, as I have identified certain issues with how it is presented.

Response: Thank you for pointing this detail. We are aware of the FAIR guidelines and, in our opinion, they were adhered in this study. While we acknowledge the current trend in data sharing and the EU's commitment to FAIR and open data policies, we opted for a controlled access approach to data sharing (only by request), as outlined in the "Data Availability Statement". This decision was made to ensure the protection of sensitive participant information, particularly given the collection of sociodemographic, employment, and personal data. Had this study followed a different design, (e.g., randomized controlled trial), we would have included the data as a supplement or provided a link. However, due to the specific nature of our research, additional security measures and ethical considerations were paramount.

The reported number of injuries and health problems is relatively low. Given the reduced sample size, employing data visualization techniques, such as a frequency count table, would provide a clearer context of the injuries. It should be possible to precisely identify the setups that are most burdensome.

Response: We have carefully considered this concern, and we acknowledge the importance of providing a comprehensive overview of injuries within the context of fitness instructor characteristics. To address this, we have included Table 1 (Fitness instructors’ personal and occupational characteristics by the reported and unreported injuries), Table 2 (Fitness instructors reported health problems characterization), and Figure 2 (Reported health problems type and localization), which collectively offer a detailed snapshot of the injuries and associated characteristics. Additionally, we have provided a correlation table analysis (Table 3) that encompasses all variables, facilitating the identification of significant relationships between different factors. However, if there is a need for more explicit data or tabulation on this matter, please do not hesitate to inform us, and we will make the necessary adjustments accordingly.

The methodology for converting some variables for model analysis is unclear. The categorization of fitness activities as either work or individual sports activities, and the summation of loads for legs as a factor, are confusing. Additionally, without a contingency table, it is difficult to ascertain the amount of time these physically active individuals spend as fitness instructors.

Response: Thank you for bringing up these concerns. We acknowledge the need for clarity regarding the methodology for converting variables for model analysis. As referred in the text “Prior to examining the associations between variables, some categories were collapsed and renamed to derive stable models in logistic regression analyses. This decision was informed by previous analysis of response frequency of items and contextual factors.” We evaluated the frequencies of each item and determined which categorizations could be merged or renamed, particularly where there were low responses to specific items. It was crucial to ensure that these adjustments were appropriate for the personal-occupation-injuries context. To provide further clarification, for socio-demographic and personal-related items, it was adjusted: age (new categorization: 18-29 years; 30-39 years; 40-50 years); body weight (new categorization: less than 50 kilograms; 50-74 kilograms; 75-100 or more than 100 kilograms); educational level (new categorization: Undergraduate; Bachelor; Master or PhD; Other); sports background weekly practice (new categorization: once per week; 2-3 times per week; more than 3 times per week; no sport or physical activity ); sports background time spent (new categorization: 30-59 minutes; 60-89 minutes; 90-120 or more minutes; no sport or physical activity); and sports background competitive level (new categorization: Recreational; Beginner; Competitive; no sport or physical activity). For the occupational related items, it was altered: years working as fitness instructor (new categorization: less than 1 year-9 years; 10-20 or more years); weekly days working as fitness instructor (new categorization: Occasional [1-6 days]; Everyday [7 days]); daily hours working as fitness instructor (new categorization: Partial [less than 1-6 hours]; Full-time [7-10 hours]; Extra-time [more than 10 hours]); main activity duration (new categorization: less than 1 hour; 1-2 hours; more than 2 hours); and working at other professions (new categorization: Yes [1 or 2]; No). For the health problems related items, it was changed: number of health problems (new categorization: 1; 2; 3; 4; 5 or more); and time returning to work (new categorization: less than 1 week; 1-2 weeks; 3-4 weeks; 1-3 months; 4-6 months; 7-12 or more months). Regarding the categorization of fitness activities as either work or individual sports activities, and the summation of loads for legs as a factor, we are sorry but we did not understand exactly what the issue is. Can you please provide the exact localization of the issue? We believe there may be a misunderstanding regarding the information presented in Table 1. To clarify, when data is presented in the table relating to sport or physical activity, we are referring to activities undertaken outside their work as fitness instructors. For example, individuals may engage in recreational or competitive sports/physical activities after their working hours. For the amount of time these physically active individuals spend as fitness instructors, we believe that the data presented in rows 11, 12, and 14, second column of the Table 1, provide sufficient information to consider their workload. However, if you require additional information to further elucidate this or other issues, please let us know.

In conclusion, I would recommend employing clustering to create specific profiles in such analyses. The overlap in data, due to the difficulty in distinguishing between work and sport activities within this particular group and survey, complicates the analysis. Furthermore, the question of whether instructors in mind and body classes, who are less likely to report injuries compared to those who constantly train with clients or personal trainers, are included in the injury-reporting group remains unanswered.

Response: Only one instructor reported an injury, specifically listing mind and body classes as their main activity. Thus, it becomes challenging to draw definitive conclusions regarding the distinct factors between various occupational activities.

The use of classical statistics in reporting injury-related outcomes in empirical studies can be challenging, leading to numerous uncertainties and raising many questions. While the discussion section is well-executed, there might be an inherent bias in your results. I recommend a revision of your work or a request for clarification on why my suggested approach may be incorrect—although this perspective is based on the presentation of your text. Therefore, at a minimum, further explanations should be provided. However, I strongly suggest considering a different descriptive statistical approach, employing contingency tables or clustering methods. Your statistical software should support these options.

Response: Thank you for your value feedback. We hope that with our explanations many of the issues found were overpassed.

Reviewer 2 Report

Comments and Suggestions for Authors

Thank you for your manuscript. Your topic is interesting; however, the article needs some improvements. Please see my comments below…

ABSTRACT

Please, define the study aim…

P1L15. “A questionnaire covering sociodemographic, occupational, health-related items was administered.” – In which population?

P1L17. “Statistical analyses, including Mann-Whitney U and chi-square tests, Spearman’s rho correlations, and logistic regressions, were conducted.” – Comparing and correlating what?

INTRODUCTION

P1L34. “Of the people living in Europe, 63.1 million are health or fitness club members, which represents an annual increase of 12.3% [2].” – This sentence does not make sense...

P1L36. “…(an annual increase of 0.5% and 66%, respectively)…” – Respectively what? It does not make sense...

P1L37. “It is estimated that almost 750,000 persons work within this industry in Europe, USA or Australia [3].” – This reference studied Norwegian group fitness instructors… How do they have this data?

P2L51. “Most of the time, fitness instructor´s find themselves having to carry out the exercises themselves while teaching, and they may work many hours, days and nights, weekends, holidays, and may even have to travel to different gyms or to clients’ homes to teach classes or to conduct personal training sessions, which can lead to a variety of musculoskeletal injuries [1, 5].” – Examples of injuries are needed... what do previous studies say? This part of the Introduction section should be more elaborate.... Overuse and overtraining are important concepts…

Although redundant, I believe it is important to mention in the study aim that it was conducted on the Portuguese population...

MATERIAL AND METHODS

Is there no information from participants regarding the region where they work and live? Are they from the coast or inland? From the north or south of Portugal? Do they represent the population of Portuguese fitness instructors?

P3L99. “The exclusion criteria from the study were: inadequate Portuguese language skills; be under 18 years old; do not hold a valid professional title of exercise technician; and currently not working as fitness instructor.” – This sentence is redundant…

P3L99. “Occupational and sociodemographic-related items. This questionnaire phase contained questions regarding age, sex, height, weight, education level, sports background, years as fitness instructors, weekly instruction, workload, main instruction activity, workplace, and other professional activities.” – I only count 12 items. Above, 18 are mentioned…

P4L137. “After checking for the data normal distribution, independent two samples non-parametric Mann-Whitney U (ordinal data) and chi-square tests (nominal data) were carried out.” – Comparing what?

P4L137. “...0 to 0.20 — Negligible; 0.21 to 0.40 — Weak; 0.41 to 0.60 — Moderate; 0.61 to 0.80 — Strong; and 0.81 to 1.00 — Very strong.” – What about negative values?

RESULTS

P4L160. “From the estimated 9061 fitness instructors actively working in Portugal, only 1462 could be reached...” – What are the reasons why the rest of instructors are not available?

P9L194. “In comparison with the fitness instructors that reported having an occupational health problem versus those who unreported in the sex (p=0.012), years as fitness instructors (p=0.001), weekly days working (p=0.039), and daily hours working (p=0.013) items significant statistical differences were found.” – It will be important to mention in which direction these differences are...

DISCUSSION

I think the first paragraph should be more objective... As it stands, it is not of great interest for the Discussion section…

P1L259. “…lower back (13.6%), hip (10.2%), lumbar spine...” – What is the difference between "lower back” and “lumbar spine”?

It is not expected that a discussion will present data, unless they are essential to understand the context...

P1L277. “An occupational injury may be defined as one that is caused, or made worse, by exposure at work [25].” – This should be in the Introduction section…

What is discussed in the 3rd paragraph?

P2L288. “Taking into consideration the disorders found in this and other international studies with fitness instructors samples, most of them can be classified as overuse injuries, resulting from repetitive force applied to a tissue [1, 17].” – How do you achieve this conclusion? Based on what data? This is speculating...

5º paragraph – 1 subject reported vocal problems (1 in 212). Is it significant?

6º paragraph – 5 subjects reported psychological disorders (5 in 212). Is it significant? What do previous studies say regarding the prevalence of psychological disorders?

There is no discussion regarding the prevalence of health problems in this sample (27.8% reported that). What do previous studies say regarding this?

CONCLUSION

P4L416. “…since 416 it can be considered a hazardous profession.” – What are the conditions for a profession to be considered hazardous? References?

Comments on the Quality of English Language

No comments...

Author Response

ABSTRACT

Please, define the study aim…

P1L15. “A questionnaire covering sociodemographic, occupational, health-related items was administered.” – In which population?

P1L17. “Statistical analyses, including Mann-Whitney U and chi-square tests, Spearman’s rho correlations, and logistic regressions, were conducted.” – Comparing and correlating what?

Response: Indeed, we acknowledge that some information is missing from the Abstract section. However, the MDPI guidelines impose a word limit of 200. As you may have noticed, our study involved several data analyses and yielded significant results that needed to be included. We made every effort to create the most informative abstract within the word limit. Currently, the abstract contains 200 words, so regrettably, we cannot add much more information. Nevertheless, we made the effort to include at least the study aim. It is worth noting that while certain details you requested are absent from the abstract, they are presented in the main text.

INTRODUCTION

P1L34. “Of the people living in Europe, 63.1 million are health or fitness club members, which represents an annual increase of 12.3% [2].” – This sentence does not make sense...

Response: You are correct. That was indeed an error in writing English. The sentence has been improved. As you also noticed, there were more misspellings encountered throughout the text. Therefore, the manuscript was reviewed for English language errors.

P1L36. “…(an annual increase of 0.5% and 66%, respectively)…” – Respectively what? It does not make sense...

Response: The sentence highlights the anticipated growth in both the number of fitness clubs and their revenues. Specifically, there is a projected increase of 0.5% in new fitness clubs and a 66% increase in revenues. However, we agree with your comment, and the sentence has been revised for clarity and accuracy.

P1L37. “It is estimated that almost 750,000 persons work within this industry in Europe, USA or Australia [3].” – This reference studied Norwegian group fitness instructors… How do they have this data?

Response: Yes, this reference is from a Norwegian group. In that study, they reported the data from two studies: EHFA. (2012). European health and fitness association annual report 2012 (pp. 1–28). Brussels, Belgium; and Fitness Australia. (2012). The Australian fitness industry report 2012 (pp. 1–28). New South Wales, Australia. Unfortunately, we were unable to access the data provided in these two individual studies. Therefore, we chose to cite the Norwegian study. However, if you have more precise data or studies regarding the estimated number of persons working in the fitness industry, please let us know.

P2L51. “Most of the time, fitness instructor´s find themselves having to carry out the exercises themselves while teaching, and they may work many hours, days and nights, weekends, holidays, and may even have to travel to different gyms or to clients’ homes to teach classes or to conduct personal training sessions, which can lead to a variety of musculoskeletal injuries [1, 5].” – Examples of injuries are needed... what do previous studies say? This part of the Introduction section should be more elaborate.... Overuse and overtraining are important concepts…

Response: We acknowledge that additional information could be incorporated into the introduction section, particularly concerning previous studies and related injuries. However, much of this data was already addressed in the discussion section, where we extensively discussed and compared it with our findings. To avoid redundancy, we chose to exclusively explore these themes in the discussion section. Similarly, the concepts of overuse and overtraining were thoroughly examined, and the data were consolidated in the fourth paragraph of the discussion section. So, while we recognize that the introduction section could be expanded upon, we believe that for the sake of avoiding repetition throughout the manuscript, this information should be exclusively addressed in the discussion section.

Although redundant, I believe it is important to mention in the study aim that it was conducted on the Portuguese population...

Response: As requested, we added the information to the text.

MATERIAL AND METHODS

Is there no information from participants regarding the region where they work and live? Are they from the coast or inland? From the north or south of Portugal? Do they represent the population of Portuguese fitness instructors?

Response: Through our established mode of contact, we were able to reach all regions of Portugal. However, as outlined in the limitations section, they do not represent the population of Portuguese fitness instructors, because the goal of the 369 responses was not achieved.

P3L99. “The exclusion criteria from the study were: inadequate Portuguese language skills; be under 18 years old; do not hold a valid professional title of exercise technician; and currently not working as fitness instructor.” – This sentence is redundant…

Response: We agree. The text as retrieved.

P3L99. “Occupational and sociodemographic-related items. This questionnaire phase contained questions regarding age, sex, height, weight, education level, sports background, years as fitness instructors, weekly instruction, workload, main instruction activity, workplace, and other professional activities.” – I only count 12 items. Above, 18 are mentioned…

Response: We understand the raised issue. Indeed, this is a result of misspelling between items and questions. When we initially constructed the questionnaire, we created a guideline with the themes (items) with corresponding questions to gain deeper insights into the experiences of fitness instructors. Consequently, some items included multiple questions. For instance, the workload item comprised three questions. This discrepancy led to the mismatch in numbers. To address this and fulfill your request, we have revised the text accordingly.

P4L137. “After checking for the data normal distribution, independent two samples non-parametric Mann-Whitney U (ordinal data) and chi-square tests (nominal data) were carried out.” – Comparing what?

Response: We believed that the statistical comparisons were clear. However, we acknowledge that they could be improved for clarity. Therefore, text was added.

P4L137. “...0 to 0.20 — Negligible; 0.21 to 0.40 — Weak; 0.41 to 0.60 — Moderate; 0.61 to 0.80 — Strong; and 0.81 to 1.00 — Very strong.” – What about negative values?

Response: We think that is redundant to also mention the negative values, because they do not alter the strength of the correlation, only its direction.

RESULTS

P4L160. “From the estimated 9061 fitness instructors actively working in Portugal, only 1462 could be reached...” – What are the reasons why the rest of instructors are not available?

Response: To maximize outreach to fitness instructors, we engaged national associations, formal schools, training facilities, large and small fitness centers, and social media platforms for disseminating the questionnaire. Out of these efforts, 1462 fitness instructors were confirmed. The unavailability of many others may be attributed to factors inherent to the profession and the country (Portugal) context. Firstly, unlike some countries, there is no requirement for fitness instructors in Portugal to be affiliated with national professional associations. Secondly, numerous fitness instructors operate independently without association with any fitness centers or “brands”. For that reason, reaching a more expressive sample was challenging.

P9L194. “In comparison with the fitness instructors that reported having an occupational health problem versus those who unreported in the sex (p=0.012), years as fitness instructors (p=0.001), weekly days working (p=0.039), and daily hours working (p=0.013) items significant statistical differences were found.” – It will be important to mention in which direction these differences are...

Response: While we acknowledge the concern raised, we believe that this information is more appropriately presented in Table 1, rather than being reiterated in the text.

DISCUSSION

I think the first paragraph should be more objective... As it stands, it is not of great interest for the Discussion section…

Response: We understand your point. The purpose of the first paragraph was to introduce the discussion section and provide a contextual overview of how the discussion will be structured. However, if you believe that additional information should be included or if the paragraph should be retrieved, please inform us accordingly.

P1L259. “…lower back (13.6%), hip (10.2%), lumbar spine...” – What is the difference between "lower back” and “lumbar spine”?

Response: The "lower back" refers to the area below the ribcage and above the hips, while the "lumbar spine" specifically denotes the lumbar portion of the vertebral column. Differentiating between the two is important due to variations in injury patterns; for instance, discopathies are typically associated with the lumbar spine, whereas muscle tears (like those in the quadratus lumborum) are more commonly linked to the lower back.

It is not expected that a discussion will present data, unless they are essential to understand the context...

Response: We acknowledge your comment. Whenever we presented our data in the discussion section, it served one of two purposes: 1) to introduce the theme to be discussed in each paragraph; 2) to provide context for readers to compare it with the overall data or other similar studies/data. Thus, the importance of maintaining the data in the discussion section.

P1L277. “An occupational injury may be defined as one that is caused, or made worse, by exposure at work [25].” – This should be in the Introduction section…

What is discussed in the 3rd paragraph?

P4L416. “…since 416 it can be considered a hazardous profession.” – What are the conditions for a profession to be considered hazardous? References?

Response: We will address these three issues together in one response, as they fall under the same overarching concern. The reason for not including the 3rd paragraph in the introduction section is that, without the data, we cannot confirm whether fitness instructors meet the conditions to be considered a hazardous profession or not. Additionally, most of the information gathered in the paragraph consists of the conditions and factors to considered a hazardous profession in Portugal (particularly those relevant to the fitness instructor profession). We consider this information less pertinent for the reader in the introductory section of the manuscript. Instead, we believe it would be more appropriate to address this topic in the discussion section, where we analyze the health issues, the profession, and the contextual factors specific to the country. Regarding your comment on the conclusion section, we agree that, as investigators, we cannot definitively determine what constitutes a hazardous profession. This determination is guided by international guidelines, followed by national or institutional regulations. Each country's government collects epidemiological data on professions and evaluates if they meet the criteria for hazardous classification, considering factors such as injury patterns, risk factors, and contextual factors (personal, occupational, sociocultural). Therefore, after analyzing our data, we referred to international guidelines (such as, those from the International Labour Organization) to identify if any of the health problems found met the eligibility criteria. Additionally, we searched for information specific to Portugal to understand the conditions for classifying a profession as hazardous (Perista, H., Cardoso, A., Carrilho, P., Nunes, J., and E. Quintal, Inquérito às Condições de Trabalho em Portugal Continental: Trabalhadores/as. Autoridade para as Condições do Trabalho, 2016). We acknowledge that defining what constitutes a hazardous profession is beyond our scope, and we have adjusted the conclusion accordingly.

P2L288. “Taking into consideration the disorders found in this and other international studies with fitness instructors samples, most of them can be classified as overuse injuries, resulting from repetitive force applied to a tissue [1, 17].” – How do you achieve this conclusion? Based on what data? This is speculating...

Response: We understand when you affirm that is speculating because we can only analyze the data and observe the injury patterns. We did not perform specific tests to affirm with certainty whether they were overuse or acute. Nevertheless, we think that our statement is not totally false. Overuse injuries typically arise from the gradual accumulation of repetitive low-energy transfer to tissues over time (Bahr, R., Clarsen, B., Derman, W., Dvorak, J., Emery, C.A., Finch, C.F., Hägglund, M., Junge, A., Kemp, S., Khan, K.M., Marshall, S.W., Meeuwisse, W., Mountjoy, M., Orchard, J.W., Pluim, B., Quarrie, K.L., Reider, B., Schwellnus, M., Soligard, T., Stokes, K.A., Timpka, T., Verhagen, E., Bindra, A., Budgett, R., Engebretsen, L., Erdener, U., and K. Chamari, International Olympic Committee consensus statement: methods for recording and reporting of epidemiological data on injury and illness in sports 2020 (including the STROBE extension for sports injury and illness surveillance (STROBE-SIIS)). Orthopaedic journal of sports medicine, 2020. 8(2): p. 2325967120902908). As highlighted in the Merati et al. (2021) study, fitness instructors are more predisposed to overuse musculoskeletal injuries due to the inherent demands of their profession, characterized by repetitive movements and prolonged periods of physical activity. Furthermore, Bratland-Sanda et al. (2015) found a higher prevalence of overuse injuries compared to acute injuries among instructors, particularly when considering workload factors. Our study's results and subsequent discussion sections corroborate these observed patterns.

5º paragraph – 1 subject reported vocal problems (1 in 212). Is it significant?

6º paragraph – 5 subjects reported psychological disorders (5 in 212). Is it significant? What do previous studies say regarding the prevalence of psychological disorders?

Response: We will address these two issues together in one response, as they pertain to the same overarching concern. In our study, injury data can be categorized into musculoskeletal injuries (such as tendon, muscle, joint, …) and other health disorders (such as respiratory infections, vocal problems, and psychological disorders). The disparity between these two injury types was evident, with musculoskeletal injuries being more prevalent. However, it is crucial to address both situations, as other health disorders have also been documented in national and international studies (e.g., Merati, G., Bonato, M., Agnello, L., Grevers, D., Gunga, H.C., Mendt, S., and M.A. Maggioni, Occupational Disorders, Daily Workload, and Fitness Levels Among Fitness and Swimming Instructors. Frontiers in public health, 2021. 9: p. 666019; Gjestvang, C., Bratland-Sanda, S., and T.F. Mathisen, Compulsive exercise and mental health challenges in fitness instructors; presence and interactions. Journal of eating disorders, 2021. 9(1): p. 1-8; Venkatraman, A., Fujiki, R.B., and M.P. Sivasankar, A review of factors associated with voice problems in the fitness instructor population. Journal of Voice, 2021; Vieira, I., Esteves, D., Ramos, L., Simões, V., and S. Franco, Quality of life of fitness professionals in Portugal: Comparative and correlation study. Frontiers in Psychology, 2022. 13: p. 958063). Additionally, as outlined in the limitations section, we were only able to explore the most significant health problems, and based on the expected data, other health disorders were not anticipated to be that prevalent (13.6% combined). Furthermore, it is essential to consider not just the absolute number of cases but also the percentages. For instance, psychological disorders represented 8.5%, which is higher than the prevalence of more commonly expected injuries such as cartilage, meniscal, and fascial injuries combined. Similarly, the incidence of vocal and respiratory infections should not be ignored. As referred in the discussion section, these two health problems are often interconnected and, when analyzed together, their collective impact could be as significant as ligamentous injuries (which are typically more expected given the demands of the profession). Therefore, in our view, and based on our data and findings from other international studies, it is essential to focus on and understand the underlying reasons for these less common injuries and not solely concentrate on the predominant ones (even though the majority of the discussion section was dedicated to musculoskeletal injuries).

There is no discussion regarding the prevalence of health problems in this sample (27.8% reported that). What do previous studies say regarding this?

Response: You are correct. The overall prevalence of health problems was not thoroughly discussed. This was because, when we attempted to compare our findings with those of other studies, we encountered variations in the overall prevalence due to different approaches (e.g., Merati, G., Bonato, M., Agnello, L., Grevers, D., Gunga, H.C., Mendt, S., and M.A. Maggioni, Occupational Disorders, Daily Workload, and Fitness Levels Among Fitness and Swimming Instructors. Frontiers in public health, 2021. 9: p. 666019; Bratland-Sanda, S., Sundgot-Borgen, J., and G. Myklebust, Injuries and musculoskeletal pain among Norwegian group fitness instructors. European journal of sport science, 2015. 15(8): p. 784-792; du Toit, V. and R. Smith, Survey of the effects of aerobic dance on the lower extremity in aerobic instructors. Journal of the American Podiatric Medical Association, 2001. 91(10): p. 528-532; Romaine, L.J., Davis, S. E., Casebolt, K., and K.A. Harrison, Incidence of injury in kickboxing participation. The Journal of Strength & Conditioning Research, 2003. 17(3): p. 580-586). Therefore, it would not have been methodologically equitable to compare such results (Dickersin, K. (2002). Systematic reviews in epidemiology: Why are we so far behind? International Journal of Epidemiology, 31(1), 6e12). As a result, we chose to concentrate more on the prevalence of individual injuries and their patterns.

Round 2

Reviewer 1 Report

Comments and Suggestions for Authors

I am grateful that authors provides me with detailed anwsers. I appriciate that at some points they explained in detail their standing as confrontation to my concerns instead of just abiding to it. 

Privacy of data is reasonable explanation. Making it anonymous may not be enough, and encoding data will make it not reprudictable anyway, without sofisticated methods.

I am glad that correlation put some more insight into the result section.

Thera are visiable changes in the manuscript, it is more clear now. As I was in favor of your progress in the first run, now I am in favor of accepting this paper.

Author Response

I am grateful that authors provides me with detailed anwsers. I appriciate that at some points they explained in detail their standing as confrontation to my concerns instead of just abiding to it. 

Privacy of data is reasonable explanation. Making it anonymous may not be enough, and encoding data will make it not reprudictable anyway, without sofisticated methods.

I am glad that correlation put some more insight into the result section.

Thera are visiable changes in the manuscript, it is more clear now. As I was in favor of your progress in the first run, now I am in favor of accepting this paper.

Response: We want to express our gratitude for your dedication and assistance in reviewing our manuscript. Your insightful scientific feedback has greatly contributed to the improvement of our work.

Reviewer 2 Report

Comments and Suggestions for Authors

You improved the article however there are points that should be revised...

MATERIAL AND METHODS

Is there no information from participants regarding the region where they work and live? Are they from the coast or inland? From the north or south of Portugal? Do they represent the population of Portuguese fitness instructors?

Your Response: Through our established mode of contact, we were able to reach all regions of Portugal. However, as outlined in the limitations section, they do not represent the population of Portuguese fitness instructors, because the goal of the 369 responses was not achieved.

My comment: Your answer must be included in the article...

RESULTS

P9L194. “In comparison with the fitness instructors that reported having an occupational health problem versus those who unreported in the sex (p=0.012), years as fitness instructors (p=0.001), weekly days working (p=0.039), and daily hours working (p=0.013) items significant statistical differences were found.” – It will be important to mention in which direction these differences are...

Your Response: While we acknowledge the concern raised, we believe that this information is more appropriately presented in Table 1, rather than being reiterated in the text.

My comment: In order to have a clearer text, I don't think so...

5º paragraph – 1 subject reported vocal problems (1 in 212). Is it significant?

6º paragraph – 5 subjects reported psychological disorders (5 in 212). Is it significant? What do previous studies say regarding the prevalence of psychological disorders?

Your Response: We will address these two issues together in one response, as they pertain to the same overarching concern. In our study, injury data can be categorized into musculoskeletal injuries (such as tendon, muscle, joint, …) and other health disorders (such as respiratory infections, vocal problems, and psychological disorders). The disparity between these two injury types was evident, with musculoskeletal injuries being more prevalent. However, it is crucial to address both situations, as other health disorders have also been documented in national and international studies (e.g., Merati, G., Bonato, M., Agnello, L., Grevers, D., Gunga, H.C., Mendt, S., and M.A. Maggioni, Occupational Disorders, Daily Workload, and Fitness Levels Among Fitness and Swimming Instructors. Frontiers in public health, 2021. 9: p. 666019; Gjestvang, C., Bratland-Sanda, S., and T.F. Mathisen, Compulsive exercise and mental health challenges in fitness instructors; presence and interactions. Journal of eating disorders, 2021. 9(1): p. 1-8; Venkatraman, A., Fujiki, R.B., and M.P. Sivasankar, A review of factors associated with voice problems in the fitness instructor population. Journal of Voice, 2021; Vieira, I., Esteves, D., Ramos, L., Simões, V., and S. Franco, Quality of life of fitness professionals in Portugal: Comparative and correlation study. Frontiers in Psychology, 2022. 13: p. 958063). Additionally, as outlined in the limitations section, we were only able to explore the most significant health problems, and based on the expected data, other health disorders were not anticipated to be that prevalent (13.6% combined). Furthermore, it is essential to consider not just the absolute number of cases but also the percentages. For instance, psychological disorders represented 8.5%, which is higher than the prevalence of more commonly expected injuries such as cartilage, meniscal, and fascial injuries combined. Similarly, the incidence of vocal and respiratory infections should not be ignored. As referred in the discussion section, these two health problems are often interconnected and, when analyzed together, their collective impact could be as significant as ligamentous injuries (which are typically more expected given the demands of the profession). Therefore, in our view, and based on our data and findings from other international studies, it is essential to focus on and understand the underlying reasons for these less common injuries and not solely concentrate on the predominant ones (even though the majority of the discussion section was dedicated to musculoskeletal injuries).There is no discussion regarding the prevalence of health problems in this sample (27.8% reported that). What do previous studies say regarding this?

My comment: Let's be objective and scientific... First, 5 subjects reported psychological disorders (5 in 212 is not 8.5%)… second, I believe that you are more focused on other studies and beliefs than your data. In agreement, I reiterate the need to reformulate this paragraph...

Author Response

Is there no information from participants regarding the region where they work and live? Are they from the coast or inland? From the north or south of Portugal? Do they represent the population of Portuguese fitness instructors?

Your Response: Through our established mode of contact, we were able to reach all regions of Portugal. However, as outlined in the limitations section, they do not represent the population of Portuguese fitness instructors, because the goal of the 369 responses was not achieved.

My comment: Your answer must be included in the article...

Response: As requested, we added to the manuscript the information provided.

P9L194. “In comparison with the fitness instructors that reported having an occupational health problem versus those who unreported in the sex (p=0.012), years as fitness instructors (p=0.001), weekly days working (p=0.039), and daily hours working (p=0.013) items significant statistical differences were found.” – It will be important to mention in which direction these differences are...

Your Response: While we acknowledge the concern raised, we believe that this information is more appropriately presented in Table 1, rather than being reiterated in the text.

My comment: In order to have a clearer text, I don't think so...

Response: To meet the request, text has been added to highlight the differences direction

5º paragraph – 1 subject reported vocal problems (1 in 212). Is it significant?

6º paragraph – 5 subjects reported psychological disorders (5 in 212). Is it significant? What do previous studies say regarding the prevalence of psychological disorders?

Your Response: We will address these two issues together in one response, as they pertain to the same overarching concern. In our study, injury data can be categorized into musculoskeletal injuries (such as tendon, muscle, joint, …) and other health disorders (such as respiratory infections, vocal problems, and psychological disorders). The disparity between these two injury types was evident, with musculoskeletal injuries being more prevalent. However, it is crucial to address both situations, as other health disorders have also been documented in national and international studies (e.g., Merati, G., Bonato, M., Agnello, L., Grevers, D., Gunga, H.C., Mendt, S., and M.A. Maggioni, Occupational Disorders, Daily Workload, and Fitness Levels Among Fitness and Swimming Instructors. Frontiers in public health, 2021. 9: p. 666019; Gjestvang, C., Bratland-Sanda, S., and T.F. Mathisen, Compulsive exercise and mental health challenges in fitness instructors; presence and interactions. Journal of eating disorders, 2021. 9(1): p. 1-8; Venkatraman, A., Fujiki, R.B., and M.P. Sivasankar, A review of factors associated with voice problems in the fitness instructor population. Journal of Voice, 2021; Vieira, I., Esteves, D., Ramos, L., Simões, V., and S. Franco, Quality of life of fitness professionals in Portugal: Comparative and correlation study. Frontiers in Psychology, 2022. 13: p. 958063). Additionally, as outlined in the limitations section, we were only able to explore the most significant health problems, and based on the expected data, other health disorders were not anticipated to be that prevalent (13.6% combined). Furthermore, it is essential to consider not just the absolute number of cases but also the percentages. For instance, psychological disorders represented 8.5%, which is higher than the prevalence of more commonly expected injuries such as cartilage, meniscal, and fascial injuries combined. Similarly, the incidence of vocal and respiratory infections should not be ignored. As referred in the discussion section, these two health problems are often interconnected and, when analyzed together, their collective impact could be as significant as ligamentous injuries (which are typically more expected given the demands of the profession). Therefore, in our view, and based on our data and findings from other international studies, it is essential to focus on and understand the underlying reasons for these less common injuries and not solely concentrate on the predominant ones (even though the majority of the discussion section was dedicated to musculoskeletal injuries).

My comment: Let's be objective and scientific... First, 5 subjects reported psychological disorders (5 in 212 is not 8.5%)… second, I believe that you are more focused on other studies and beliefs than your data. In agreement, I reiterate the need to reformulate this paragraph...

Response: For the first part of your comment the text was reviewed as requested. Regarding the second part of your comment, we respectfully disagree with the assertion that we prioritize other studies and beliefs over our data. As one of your comments in the first round, it is necessary to consider findings from other studies to better understand our results. This is precisely what we undertook, comparing data from other studies with our findings and analyzing potential reasons for the observed patterns. As evidenced (other of your prior comments was to elucidate the rationale behind presenting our results in the discussion section), the discussion revolved around our results, ensuring that they are the primary focus rather than being influenced by other studies. Therefore, we think it is crucial to continue to reference other studies, in order to better interpret and understand our data.